



# A susceptibility-based rainfall threshold approach for landslide occurrence

Elise Monsieurs[1,2,3], Olivier Dewitte[1], Alain Demoulin[2,3]

[1] Royal Museum for Central Africa, Leuvensesteenweg 13, 3080 Tervuren, Belgium
[2]Department of Geography, University of Liège, Clos Mercator 3, 4000 Liège, Belgium
[3]F.R.S.-FNRS, Egmontstraat 5, 1000 Brussel, Belgium

*Correspondence to*: Elise Monsieurs (elise.monsieurs@africamuseum.be)

**Abstract.** Rainfall threshold determination is a pressing issue in the landslide scientific community. While main improvements have been made towards more reproducible techniques for the identification of triggering conditions for
landsliding, the now well-established rainfall intensity or event – duration thresholds for landsliding suffer from several limitations. Here, we propose a new approach of the frequentist method for threshold definition based on satellite-derived antecedent rainfall estimates directly coupled with landslide susceptibility data. Adopting a bootstrap statistical technique for the identification of threshold uncertainties at different exceedance probability levels, it results in thresholds expressed as $AR = (\alpha \pm \Delta\alpha) * S^{(\beta \pm \Delta\beta)}$, where $AR$ is antecedent rainfall (mm), $S$ is landslide susceptibility, α and β are scaling
parameters, and Δα and Δβ are their uncertainties. The main improvements of this approach consist in: (1) using spatially continuous satellite rainfall data, (2) giving equal weight to rainfall characteristics and ground susceptibility factors in the definition of spatially varying rainfall thresholds, (3) proposing an exponential antecedent rainfall function that involves past daily rainfall in the exponent to account for the different lasting effect of large versus small rainfall, (4) quantitatively exploiting the lower parts of the cloud of data points, most meaningful for threshold estimation, and (5) merging the
uncertainty on landslide date with the fit uncertainty in a single error estimation. We apply our approach in the western branch of the East African Rift based on landslides that occurred between 2001 and 2018, satellite rainfall estimates from the Tropical Rainfall Measurement Mission Multi-satellite Precipitation Analysis (TMPA 3B42 RT), and the continental-scale map of landslide susceptibility of Broeckx et al. (2018) and provide first regional rainfall thresholds for landsliding in tropical Africa.

## 1 Introduction

Rainfall is widely recognized as an important trigger for landslides (Sidle and Bogaard, 2016), posing an increased threat to people and economies worldwide under climate change conditions (Gariano and Guzzetti, 2016). Rainfall thresholds, defined as the best separators for triggering and non-triggering known rainfall conditions (Crozier, 1997), are the most used instrument in landslide hazard assessment and early warning tools (Segoni et al., 2018). Whereas physically-based models





require detailed geotechnical, hydrological, and environmental parameters, which is achievable only on hillslope to small-basin scale, the empirical approach is adopted for local to global scales (Guzzetti et al., 2007).

The most common parameters used to define empirical thresholds are the combinations of rainfall intensity - duration, rainfall event - duration, and antecedent rainfall conditions (Guzzetti et al., 2007). Standard approaches for the definition of the first two combinations of parameters are in a rise (e.g., Segoni et al., 2014; Vessia et al., 2014; Robbins, 2016; Rossi et al., 2017; Melillo et al., 2018) as substitutes for the former used subjective expert-judgement approaches (Aleotti, 2004; Brunetti et al., 2010). On the other hand, no unanimous definition of triggering antecedent rainfall (*AR*) conditions is currently achieved. This is related to the complexity of environmental factors that influence the impact of *AR* on a slope (Sidle and Bogaard, 2016), yet regrettable because of *AR* physical relation with soil shear strength and thus landslide potential (Ma et al., 2014; Hong et al., 2018). *AR* has been taken into account by combining the rainfall accumulation periods identified as most significant for landslide triggering in the study area, varying up to 120 days (Guzzetti et al., 2007). In some cases, an *AR* function convoluting rainfall over the selected period is defined with the aim of reflecting the decaying effect of rainfall on soil moisture status (e.g., Crozier, 1999; Glade et al., 2000; Capparelli and Versace, 2011; Ma et al., 2014).

Once the triggering rainfall conditions of landslides have been quantitatively described, thresholds are determined through more and more refined techniques claiming objectivity and reproducibility (Segoni et al., 2018). Because the transition between triggering and non-triggering conditions for landslides cannot be sharply devised (Berti et al., 2012; Nikolopoulos et al., 2014), statistical approaches including probabilistic and frequentist methods have abandoned a deterministic approach of the threshold definition. Probabilistic methods such as Bayesian inference (Guzzetti et al., 2007; Berti et al., 2012; Robbins, 2016) are based on relative frequencies, considering information on triggering and non-triggering rainfall conditions. Critics to this method are the biased prior and marginal probabilities related to the incompleteness of the landslide input data (Berti et al., 2012). Brunetti et al. (2010) proposed a frequentist method allowing threshold definition at different exceedance probability levels, a method improved by Peruccacci et al. (2012) for the estimation of uncertainties associated with the threshold through a bootstrap statistical technique (Gariano et al., 2015; Melillo et al., 2016, 2018; Piciullo et al., 2017). A limitation of the frequentist approach is the dependency on a large and well-spread dataset in order to attain significant results (Brunetti et al., 2010; Peruccacci et al., 2012). Other, less influential, threshold identification approaches are reviewed by Segoni et al. (2018).

Regional ground conditions, but also the progressive adjustment of landscapes to the governing climatic parameters affect the meteorological conditions required for landsliding (Ritter, 1988; Guzzetti et al., 2008; Peruccacci et al., 2012). For this reason, thresholds gain in efficiency when rainfall regimes are accounted for through rainfall normalization (e.g., Guzzetti et al., 2008; Postance et al., 2018) and when the input data are partitioned according to homogeneous predisposing ground





conditions or sliding processes (Crosta, 1998; Crosta and Frattini, 2001; Peruccacci et al., 2012; Sidle and Bogaard, 2016). Yet, to the authors' knowledge no threshold mapping involving landslide susceptibility as a proxy integrating the causative ground factors has been proposed to date beyond local-scale physically-based models (e.g., Aristizábal et al., 2015; Napolitano et al., 2016). On the other hand, landslide early warning tools aim at coupling primary landslide susceptibility

data and thresholds based on rainfall characteristics, demonstrating the importance of their combination for landslide prediction at regional to global scales (Piciullo et al., 2017; Kirschbaum and Stanley, 2018).

Though being identified as a pressing issue in the scientific community, rainfall threshold research is almost inexistent in Africa (Segoni et al., 2018) despite high levels of landslide susceptibility and hazard, especially in mountainous tropical

Africa, characterized by intense rainfall, deep weathering profiles and high demographic pressure on the environment (Aristizábal et al., 2015; Jacobs et al., 2018; Monsieurs et al., 2018a). The lack of scientific investigation in this area is most likely related to the dearth of data on timing and location of landslides (Kirschbaum and Stanley, 2018). However, the other fundamental data for threshold analysis, namely rainfall data, is globally freely available through satellite rainfall estimates (SRE) since the 1990s. Even if their use in threshold analysis remains limited (Brunetti et al., 2018; Segoni et al., 2018),

SRE have many advantages in sparsely gauged areas such as tropical Africa. A review paper by Brunetti et al. (2018) reveals that, to date, the most recurring SRE products used for research on landslide triggering conditions come from the Tropical Rainfall Measuring Mission (TRMM) (e.g., Liao et al., 2010; Kirschbaum et al., 2015; Cullen et al., 2016; Robbins, 2016; Nikolopoulos et al., 2017; Rossi et al., 2017).

The main objective of this paper is to devise an improved version of the frequentist method of rainfall threshold definition that goes beyond the sole aspect of rainfall characteristics and will be applicable in regions with limited rainfall gauge data such as, e.g., tropical Africa. Consequently, it will rely on the use of TRMM satellite rainfall data. Directly operational thresholds and threshold maps are expected from several methodological improvements regarding the definition of an elaborate $AR$ function, the integration of climatic and ground characteristics (through landslide susceptibility) into a 2D

trigger-cause graph, and a better focus on the information delivered by landslide events associated to low $AR$ values. The western branch of the East African Rift (WEAR, Fig. 1) serves as a suitable study area prone to landsliding (Maki Mateso and Dewitte, 2014; Jacobs et al., 2016; Monsieurs et al., 2018a; Nobile et al., 2018), in which recent efforts have been made to collect information on landslide occurrence (Monsieurs et al., 2018a) and validate TRMM products (Monsieurs et al., 2018b).



## 2 Setting and data

### 2.1 Landslides in the WEAR

The study area extends over ~350,000 km² in the WEAR (Fig. 1). High seismicity (Delvaux et al., 2017), intense rainfall (Monsieurs et al., 2018b), deeply weathered substrates (e.g., Moeyersons et al., 2004), and steep slopes with an elevation
range of 600 m at Lake Albert to 5109 m in the Rwenzori Mountains (Jacobs et al., 2016) are all predisposing factors rendering the area highly prone to landsliding (Maki Mateso and Dewitte, 2014; Broeckx et al., 2018; Monsieurs et al., 2018a; Nobile et al., 2018).

We updated the currently most extensive database existing over the WEAR from Monsieurs et al. (2018a), which formerly
contained information on the location and date of 143 landslide events that occurred between 1968 and 2016. New information on landslide occurrence was added through an extensive search of online media reports and to a lesser extent information from local partners. Only landslides with location accuracy better than 25 km and for which the date of occurrence is known with daily accuracy are included, Monsieurs et al. (2018a) stressing that a residual uncertainty on landslide date especially affects landslides having occurred overnight. Omitting pre-2000 events so as to adjust to the
temporal coverage of the satellite rainfall data, the updated inventory comprises a total of 174 landslide events occurred between 2001 and 2018 and located with a mean accuracy of 6.7 km. Their spatial distribution is limited in the longitude axis (Fig. 1) because of data collection constraints related to the remote and unstable security conditions (Monsieurs et al., 2017). The landslide temporal pattern shows that most of them occurred after the second rainy season from March to May, almost no landslides being reported in the following dry season (June–August) (Fig. 2). Daily rainfall distributions per
20   month are provided as Supplementary Material.

A distinction is made for landslides mapped in mining areas, counting 29 out of the 174 events. As media reports generally lack scientific background and insights into the landslide process, we discard these events because of the possibility of anthropogenic interference in their occurrence. We also acknowledge that the rest of the inventory may encompass a wide
range of landslide processes, from shallow to deep-seated landsliding (Monsieurs et al., 2018a), and that another bias in the WEAR dataset highlighted by field observations is the non-recording of many landslide events (Monsieurs et al., 2017, 2018a). Therefore we claim neither catalogue completeness nor ascertained identification of the conditions determinant for landsliding.

### 2.2 Rainfall data

Owing to the absence of a dense rain gauge network in the WEAR over the study period (Monsieurs et al., 2018b), we use SRE from the TRMM Multisatellite Precipitation Analysis 3B42 Real-Time product, version 7 (hereafter spelled TMPA-RT). While the TRMM satellite is no longer operating, the multisatellite TMPA product is continued to be produced by



combining both passive microwave and infrared sensor data (Huffman et al., 2007). TMPA-RT is available at a spatiotemporal resolution of 0.25° x 0.25° and 3 h for the period 2000 to present, over 50°N – 50°S, provided by NASA with 8 h latency. Compared to the TMPA Research Version product, TMPA-RT shows lower absolute errors and was found to overall perform better in the WEAR for higher rainfall intensities (Monsieurs et al., 2018b). Yet, average rainfall underestimations in the order of ~40% and a low probability of detecting high rainfall intensities as such have to be taken into account. We maintain TMPA-RT's native spatial resolution, while aggregating the 3-hourly data to daily resolution, in accordance with the landslide inventory temporal resolution.

### 2.3 Susceptibility data

As we want to introduce ground factors directly within the frequentist estimation of rainfall thresholds, we make use of susceptibility data as a proxy for the joint effect of ground characteristics on spatial variations of thresholds. We adopt here the landslide susceptibility model from Broeckx et al. (2018). Calibrated for all landslides regardless of type and covering the African continent at a spatial resolution of 0.0033°, this model has been produced through logistic regression based on four independent environmental factors, namely topography, lithology, peak ground acceleration and precipitation. Susceptibility is expressed as the spatial probability of landslide occurrence in each pixel. As the value of these probability estimates depends strongly on the ratio between the numbers of landslide and no-landslide pixels used in the model calibration, we stress that Broeckx et al. (2018) applied a ~4:1 L/NL ratio. Interestingly, as their susceptibility map covers the whole Africa, this model characteristic will not contribute to mar potential extrapolations of our calculated thresholds to similar analyses elsewhere in the continent. Finally, when resampling the susceptibility data to the coarser 0.25° resolution of the SRE used in the threshold analyses, we assigned to each TMPA pixel a value corresponding to the 95th percentile of the original values in order to reflect the behaviour of the most landslide-prone sub-areas within the pixel.

## 3 A novel approach of the frequentist method

### 3.1 Conceptual framework

In order to overcome the limitations of the current frequentist approach of rainfall thresholds related to, e.g., the variable definition of triggering rainfall events and non-consideration of ground conditions, we feel that the generally used rainfall characteristics (intensity-duration or cumulative rainfall event-duration) should be lumped into a single metric, thus allowing space for introducing other parameters in the frequentist analysis. This has been suggested also by Bogaard and Greco (2018), who advocate a combination of meteorological and hydrological conditions into a 'trigger-cause' framework of threshold definition where short-term rainfall intensity would represent the meteorological trigger. The main limitation thereof is however the limited availability of information about the other variable, namely the causative hydrological status of slopes. We thus propose an alternative concept where an *AR* function describes the progressive building of the landslide trigger, and the set of determining causes, mostly related to ground conditions, is accounted for by landslide susceptibility. In





assigning the triggering and causative roles to *AR* and susceptibility respectively within a renewed 2D frequentist graph, we distribute in fact the influence of hydrological conditions between trigger (by devising an elaborate *AR* function that aims to reflect the hydrology of an empirical average soil) and cause (by capturing in the susceptibility the spatial variations of hydrological conditions expected from the distribution of the causative ground factors). In this way, we obtain rainfall (*AR*)

thresholds as functions of susceptibility, which enables us to associate threshold mapping with susceptibility maps. We show below how this new approach furthermore includes the definition of a more meaningful *AR* function and the use of subsets of the landslide data set in the threshold function estimation. Analyses are performed in the R open-source software, release 3.4.3 (http://www.r-project.org). The source code is provided as Supplementary Material.

## 3.2 A new antecedent rainfall function

Though various expressions of the *AR* function have been proposed (see an overview in Capparelli and Versace, 2011), most authors calculate *AR* by convolving the time series of daily (or any other length) rainfall $r_t$ with a filter function in the form of an exponential function of time *t*, over a period empirically fixed to the preceding *n* days (e.g., Langbein et al., 1990; Crozier, 1999; Glade et al., 2000; Melillo et al., 2018):

$$AR = \sum_{t=0}^{n} e^{-a*t} * r_t \tag{1}$$

Such a function, which attempts to reflect the time-decaying effect of past rainfall on the soil water status, has two weaknesses. A first one is that, *a* being a constant, *AR* does not vary the time constant for decay of the effect on soil moisture of small versus high daily rainfall. Yet, in general and especially within the weathered material veiling the slopes of the

WEAR, one may expect that infiltration depth and, thus, residence time of the rain water in the soil increase with daily rainfall. We take this into account by introducing also daily rainfall $r_t$ in the filter function and expressing *AR* as:

$$AR = \sum_{t=0}^{n} e^{\frac{-a*t}{r_t^b}} * r_t \tag{2}$$

The *b* power of $r_t$ in the exponential function allowing us to tune the gradient of residence time between rainfall of variable amount. We empirically determined that $a = 1.2$ and $b = 1.2$ provide decay curves that comply with both a realistic contrast in residence time of different rainfall in the soil and the duration of their effect on soil moisture expected in the WEAR (e.g., McGuire et al., 2002) (Fig. 3).

As for the second weakness of the usual *AR* formulation, related to the length of the period of time to be used for *AR* calculation, we stick so far to the simplest solution of relying on expert knowledge to select it depending on the regional environmental conditions. Two observations from the landslide temporal distribution are taken into account for the choice of an appropriate accumulation period: (1) landslide frequency progressively increases during the long rainy season (hardly



interrupted by a short drier period centred on January) and peaks at its end in May, suggesting that the length of the preceding period of wet conditions indeed controls landslide frequency, and (2) the abruptly decreasing number of landslides as soon as the dry season starts in June indicates that the period of useful *AR* should not exceed a few weeks (Fig. 2). As a trade-off, we choose to calculate *AR* over a period of six weeks, or 42 days. Using such a fairly long period is also required

5 because all landslide types are included in the data set, including large-scale and deep rotational slope failures that often occur only after a long rainy period (Zêzere et al., 2005; Fuhrmann et al., 2008; Robbins, 2016). A six-week period is also consistent with studies having estimated the soil water mean residence time to about two months for two watersheds in the Mid-Appalachians of central Pennsylvania, USA (McGuire et al., 2002), and shown that the best fit between creep rate on the Parkfield segment of the San Andreas Fault (California, USA) and rainfall is obtained for a time constant of about one

10 month (Roeloffs, 2001). Finally, we note in passing that another advantage of basing *AR* on a long period of time is that the effect of rainfall events missed by the satellite TMPA-RT data due to time gaps between satellite microwave observations (Monsieurs et al., 2018b), is reduced.

### 3.3 Definition of AR thresholds for landslides

Owing to the variables we employ to construct the frequentist graph, the rainfall thresholds will be given as *AR* values in

15 function of susceptibility, or landslide-predisposing ground factors. Hereby we avoid regionalizing the input data according to individual variables such as lithology, land cover, or topography, and getting into problems of data subsetting in regions with limited data (Peruccacci et al., 2012). However, the use of these variables brings us to change the statistical way of threshold calculation, which leads to conceptual improvements in the threshold definition and might also be fruitfully applied to threshold estimation based on rainfall intensity-duration or event-duration data.

The first steps of the analysis follow the procedure devised by Brunetti et al. (2010) and Peruccacci et al. (2012). We first plot the landslide data points in the 2D *AR*-susceptibility (*S*) space (Fig. 4). Only the 145 landslides unrelated to mining and human alteration of the ground conditions are considered, from which two landslides associated with *AR* < 5 mm are further discarded, as they barely can be said to have been triggered by rainfall in such conditions. While such low *AR* values cannot

25 be ascribed to errors in the TMPA-RT rainfall estimates, due to the length of the period of *AR* calculation, they might result from gross errors of landslide location or date identification, or possibly the intrinsic evolution of hillslopes with time-dependent strength degradation of the slope material resulting in slope failures without apparent trigger (Dille et al., submitted). The retained 143 landslides occurred between 2001 and 2018 and are located in 58 different TMPA/susceptibility pixels.

30

The threshold function is then approached through a regression of *AR* against *S*. As the possible relation between the two variables was a priori unknown, we tested different functions, from which a power law fit (equivalent to a linear regression in the log-log space) in the form





$$AR = (\alpha \pm \Delta\alpha) * S^{(\beta \pm \Delta\beta)}, \tag{3}$$

appeared to work best.

The uncertainties $\Delta\alpha$ and $\Delta\beta$ associated with the $\alpha$ and $\beta$ scaling parameters are obtained by a bootstrap statistical technique
where we generate 5000 series of randomly selected events from the dataset. The parameter values and their uncertainties correspond to the mean and the standard deviation, respectively, of their 5000 estimates. Peruccacci et al. (2012) applied this technique in order to get the fit uncertainty on the estimated parameters $\alpha$ and $\beta$. Here, we first produce a derived dataset that must allow merging the fit uncertainties with those upon the data themselves into the error estimates provided by the bootstrap process. Data uncertainties relate to the accuracy of landslide location and date identification. As the mean location
accuracy of 6.7 km is much better than the ~28 km pixel size, we decided to neglect this type of uncertainty. However, the dating uncertainty is more of an issue. Uncertainty on the date most frequently arises from landslides having occurred during the night. Beyond the fact that reports do not always mention it, it is also generally unsure whether any nightly landslide has been assigned to the day before or after the night. In terms of uncertainty, this implies that a reported landslide may have occurred randomly at any time over a 36-hour period centred on the reported day. To account for this randomness, we
associate each landslide with three weighted dates, the reported day having a weight of 24/36 (~0.67) and the previous and next days each a weight of 6/36 (~0.17) corresponding to the first half of the preceding night and the second half of the following night, respectively. We then simply expand, according to the day weights, the original 143-event data set to a set of 858 derived events of same 0.17 probability (in which, for each landslide, day 0 is represented four times whereas only one occurrence of days -1 and +1 is present). The date uncertainty is therefore incorporated in the expanded data set and thus
will be also included in the $\Delta\alpha$ and $\Delta\beta$ uncertainty estimates from the bootstrapping, each bootstrap iteration randomly sampling 858 independent events out of this data set, for a probability sum of (~0.17)*858 = 143. Note that a close but less practical alternative might have consisted in using the intermediate data set of 429 (=3*143) weighted events and requiring every bootstrap iteration to randomly sample a variable number of events for a total sum of weights of 143. In order to satisfy the requirement of the frequentist method for the largest possible data set, we used the entire set of landslide events
for the calibration of the new method, leaving aside a validation of our results based on updates of the WEAR data set or on landslide sets of neighbouring regions for the near future.

Once the bootstrap procedure has yielded the parameters and uncertainties of equation (3), the residuals of the regression are calculated and subsets of their largest negative values are selected according to the exceedance probabilities of the thresholds
we want to calculate. Brunetti et al. (2010) use for this a Gaussian fit to the probability density function of the population of residuals and take the residual value $\delta$ that limits the lowest x% of this fit to define the x% exceedance probability threshold as a line parallel to the global regression line (in the $\log(AR)$-$\log(S)$ space), i.e., with an unchanged $\beta$ parameter, and simply translated toward lower $\alpha$ by a distance $\delta$ (see their figure 2). Here, we prefer to put more weight on the distribution of the data points in the lower part of the cloud of points as the most meaningful part of the data set for threshold identification.





Once the residuals have been computed, we take the subset of their x% largest negative values and regress *AR* against *S* only for the corresponding data points, obtaining a new regression line with not only a lower α but also a modified β parameter that better follows the lower limit of the cloud of points. Running through the middle of the x% lowest data points, this new curve is thus taken as the threshold curve for the (x/2)% exceedance probability. In this way, the whole data set is used to

calculate the trend that allows meaningful sampling of subsets of low-*AR* points before the emphasis is put on the subsets to get curves better reflecting the actual threshold information contained in the data set. Note also that we select a subset of actual data points to estimate the threshold, whereas the approach of Brunetti et al. (2010) relies on a Gaussian approximation of the residual distribution.

## 4 AR threshold estimates

The range of *AR* values associated with the landslide events extends from 5.7 mm to 164.4 mm for landslides that occurred in areas displaying a range of susceptibility (expressed as probabilities) from 0.38 to 0.97. As a first result, the fit to the (*AR, S*) pairs of the whole set of landslide events was expressed as (Fig. 4):

$$AR = (36.5 \pm 1.2) * S^{(-0.41 \pm 0.09)} \tag{4}$$

showing a stable solution of the bootstrap with fairly small (date + fit) uncertainties on α and β but a rather small β value

indicating a weak dependence of *AR* on *S*, confirmed by the statistical non-significance of the fitted trend. We then selected two subsets of 10% and 20% of all data points with the most negative residuals with respect to the above calculated trend in order to obtain the threshold curves for the 5% and 10% probabilities of exceedance, respectively, on which power law regressions yielded (Fig. 4):

$$AR\ (5\%) = (9.2 \pm 0.6) * S^{(-0.95 \pm 0.14)} \tag{5}$$

$$AR\ (10\%) = (13.1 \pm 0.7) * S^{(-0.66 \pm 0.15)} \tag{6}$$

A 5% exceedance probability, for instance, means that any landslide occurring in the field has a 0.05 probability of being triggered by an antecedent rainfall *AR* lower than that defined by the threshold curve, with about weighted 5% of the data points effectively lying below the curve. A first observation is that the two threshold curves present significantly higher β

values than the previously calculated general trend, thus enhancing the susceptibility-dependent gradient of *AR* threshold. Maximum β value is obtained for the lowest threshold, which targets most sharply the data points of interest, while larger subsets yield values progressively closer to that of the general trend and thus less meaningful. Again, the bootstrap-derived uncertainties are rather low, even though the β uncertainties appear slightly higher than previously, probably owing to smaller sample size and narrower range of represented *S* values. At the 5% exceedance probability, the *AR* threshold

amounts to 22 mm and 9.2 mm for susceptibilities of 0.4 and 1, respectively, making an *AR* difference of ~13 mm between weakly and highly susceptible ground conditions. In the same time, the regressions on the subsets of data are now statistically significant at the 95% confidence level with average $R^2 = 0.27$ for the 5% curve and 0.13 for the 10% curve and





with quasi all single bootstrap iterations providing significant α and β parameters for both thresholds, showing that there exists a true correlation between susceptibility and rainfall threshold as soon as one focuses on the data points really pointing to the minimum *AR* required for landsliding to start.

The threshold curves of Fig. 4 have not been extrapolated over the entire possible range of susceptibility because the relation between susceptibility lower than 0.38 and triggering *AR* conditions is uncertain as long as it cannot be empirically tested. At the continental scale, pixels with a susceptibility ≤ 0.38 are ranked in any case as low- and very low-susceptibility areas (Broeckx, oral communication). In the WEAR, the only landslides that were recorded in areas with *S* < 0.38 are all related to mining activity.

**5 Discussion**

Having proposed a new approach of the frequentist method of rainfall threshold determination for landsliding, we have tested it successfully for the WEAR despite the difficulties of the context (limited size of the landslide set, heterogeneity of the study area with respect to ground conditions, coarse spatial resolution of the rainfall data). We want now to review every new element of the method and have a look at their advantages, implications, and limitations, especially from the point of

view of the added value for the landslide scientific community.

5.1 A key point of our approach is the introduction of ground characteristics as one variable of the 2D frequentist graph, with the aim of directly associating the climatic trigger of landslides and the causative ground conditions in the threshold analysis. Indeed, in the current way of treating this problem, examining separately the effect on thresholds of various ground variables

(e.g., lithology, topography) has the drawbacks that (1) partitioning the study area on the basis of categories of any variable may entail that some subsets of data become too small for a significant analysis (Peruccacci et al., 2012) and (2) the combined effect of the variables cannot be investigated. In order to put everything at once in the frequentist graph, we thus needed to use two variables synthesizing the climatic and ground characteristics, respectively. While the climatic trigger issue was fixed by proposing a refined *AR* function (see point 5.2 below), we found that susceptibility to landsliding is an

ideal single indicator integrating all ground characteristics that significantly determine the hillslope sensitivity to rainfall accumulation. In hydrological terms, susceptibility expresses how rapidly slopes come to the point where soil infiltration and drainage capacity are no longer balanced and saturated soils become prone to landsliding. As the other side of the coin, we could cite the fact that no single raw variable is explicitly stated in this approach, and especially the soil water status of slopes. But the main point is that using susceptibility values from pre-existing studies implies to know how they were

estimated. This is not a real issue for a regional study but becomes relevant if thresholds obtained somewhere were to be transposed in other regions where the susceptibility data would have been calculated in a different way. In the frequent case that susceptibility is modelled through logistic regression for example, the probabilities that quantify susceptibility have no



absolute meaning, depending on the ratio between the landslide and no-landslide sample sizes used in the modelling. Such information should thus always be specified when susceptibility data are exploited for threshold determination.

5.2 *AR* functions are a common tool to lump daily rainfall and antecedent rainfall into a single measure. In general, they
either simply use cumulated rainfall over empirically-determined significant periods or take into account the decaying effect of rainfall on the soil water status. With respect to the intensity- or cumulated rainfall event-duration descriptions of rain characteristics, they replace the difficulty of objectively defining rainfall events by that of choosing a relevant period of meaningful antecedent rainfall and, if a filter function is used, of parameterizing it. The latter also offers a better proxy for the time-varying soil moisture content (Hong et al., 2018; Melillo et al., 2018). However, no *AR* function has so far
considered that the decay time constant is likely to increase with rainfall intensity. Here, we have applied this idea by introducing daily rainfall in the filter function of *AR* as a scaling factor of the time constant (Eq. 2). In addition to the usual virtue of this *AR* function type of assigning full weight to the rainfall of the current day, this allows a better contrast between the intensity-dependent lasting effect of different past rainfall, with more weight put on high-intensity rainfall.

Another facet of the *AR* issue is that we used remotely sensed rainfall data from the TMPA-RT products (e.g., Hong et al., 2006; Robbins, 2016). In the WEAR case, this was anyway required because the existing rain gauge network in the area is neither dense nor was installed soon enough to adequately cover the study area and period. Moreover, using TMPA-RT data is advantageous in that the information is spatially continuous (Rossi et al., 2017; Postance et al., 2018) and freely available with a global coverage in near-real time (Hong et al., 2006; Kirschbaum and Stanley, 2018). The rather coarse spatial
resolution of TMPA-RT data may also turn into an advantage because of their higher spatial representativeness compared to gauge point-observations of very local meaning in areas with pronounced topography (Marra et al., 2017; Monsieurs et al., 2018b). However, one has to cope with the typical bias of SRE, which systematically underestimate rainfall amounts with respect to ground observations (Brunetti et al., 2018; Monsieurs et al., 2018b). As stated by Brunetti et al. (2018), this does not affect the performance of threshold determination as long as the bias is spatially and temporally homogeneous, which is
to some extent the case in the WEAR. Based on the estimation by Monsieurs et al. (2018b) that average SRE underestimation amounts to ~40% in this area, we calculate an approximate first-order correction of the *AR* thresholds. For instance the calculated ~13 mm difference in 5% *AR* threshold between low- and very high-susceptibility areas of the WEAR becomes ~21 mm after correction for SRE underestimation, with corrected 5% *AR* thresholds of 36.6 mm and 15.3 mm in areas with $S = 0.4$ and 1, respectively. However, Monsieurs et al. (2018b) also highlight how SRE underestimation increases
with rainfall intensity, reaching, e.g., an average 80% for daily rainfall around 30 mm. This means that, even after correction, the thresholds, in which high daily rainfall have highest weight, are still underestimated, and thus only indicative.

5.3 Another characteristic of our approach lies in the fashion of determining thresholds by focusing on the data points with lowest AR. Though this is not quite new (Althuwaynee et al., 2015; Lainas et al., 2016; Segoni et al., 2018), it is carried out



here in a statistically rigorous manner so as to exploit the part of the data most meaningful for threshold appreciation. This methodological change was needed initially because, contrary to the obvious strong relation between rainfall intensity or event rainfall and duration (Guzzetti et al., 2007), the intuitively expected relation between ground susceptibility and rainfall threshold was not at all expressed in the data, with a largely insignificant correlation between both variables. Many reasons

potentially contribute to the noise that obscures such a relation among the ($AR$, $S$) landslide data, relating to: (i) probably chiefly, the mixing of all types of landslides in our data set (Flageollet et al., 1999; Sidle and Bogaard, 2016; Monsieurs et al., 2018a); (ii) the spatial, temporal, and rain-intensity dependent inhomogeneity of TMPA-RT underestimation, with local bias caused, e.g., by high percentages of water areas within a pixel or by topographic rainfall (Monsieurs et al., 2018b); (iii) determining factors of landsliding important in the WEAR region but not accounted for in the continental-scale prediction of

susceptibility by Broeckx et al. (2018), such as slope aspect, thickness of the weathering mantle, deforestation and other human-related factors; (iv) the occurrence of landslides in less susceptible areas of a pixel classified as highly susceptible; (v) the probability of landslides having occurred in the very first hours of a day with 24-h-long high rainfall, inducing artificially swollen *pre*-landslide $AR$. By contrast, focusing on subsets of landslides with low-$AR$ residuals leads to significant correlations and thresholds with higher β values more closely reflecting the visually outstanding lower bound of

the cloud of data points and the $AR$ threshold dependence on susceptibility. Working with independent regressions on subsets strongly reduces the data noise and thus better captures the true threshold shape. In this scheme, many of the actual landslide events associated with $AR$ much higher than the calculated threshold might be viewed as 'quasi false-positives' that, for any reason, required much more rainfall than predicted before at last occurring. Regarding false positives, it is however important to note that the landslide data set used for threshold calculation is far from complete and that a lot of landslides

occurring in remote areas are de facto unreported and may even get easily unnoticed on satellite imagery if they occurred in regions with fast vegetation regrowth, land reclamation, or in places with poor temporal satellite coverage, so there probably exist many "false false-positives", i.e., ignored true positives.

5.4 A main requirement for a widely usable method of threshold calculation is an automated threshold procedure, ideally

made available online, in order to enhance reproducibility of analysis and promote worldwide comparison of results (Segoni et al., 2018). Steps towards this goal are achieved through

(1) using TMPA-RT data, a freely available, spatially homogeneous product covering the 50°N-S latitude range: this ensures that the results of other regional analyses using the same data may be safely compared with ours. The RT (real-time) version

of the product has intentionally been preferred to the more elaborated Research Version calibrated against gauge-based GPCC rainfall data (Huffman et al., 2007) because the inhomogeneous distribution of the reference gauges worldwide, and especially in the tropics, introduces a spatially variable bias into the residual underestimation of the latter data (Monsieurs et al., 2018b);



(2) reduction of the number of adjustable parameters in the definition of the climatic characteristics leading to landsliding: here, only the constant coefficient and the exponent on daily rainfall in the filter function have to be fixed, along with the length of the period over which $AR$ is calculated. A dedicated statistical study of their best values (e.g., Stewart and McDonnell, 1991; McGuire et al., 2002) might perhaps improve somewhat those we empirically defined but, in any case, tests have shown that our formulation of $AR$ is not much sensitive to moderate changes in these values;

(3) drawing attention onto the effect on the calculated thresholds of the way the used susceptibility data have been obtained: in particular, it is possible to correct the threshold results for differences in the ratio between the landslide and no-landslide sample sizes used with the widely recognized logistic regression model of susceptibility;

(4) improving the evaluation of uncertainty: all sources of uncertainties (here, date and fit uncertainties but location uncertainty, e.g., may be treated alike) are merged into a single error estimation in a bootstrap procedure randomly taking from a weighted data set samples that have the same size in terms of sum of the weights (or probabilities) of the selected events rather than in the number of events;

(5) providing our source code as Supplementary Material.

5.5 Beside method development, this study has yielded valuable new regional information in the form of $AR$ threshold-susceptibility relations and a threshold map at 0.25° × 0.25° resolution (Fig. 5). These results are immediately usable for early warning of landslide hazard in the WEAR. Depending on the local susceptibility, thresholds at 5% exceedance probability, which we consider the best operational measure, range from $AR$ = ~15.3 mm (corrected for SRE underestimation) in the highest-susceptibility areas to 38.4 mm in the least susceptible pixels ($S$ = 0.38) having recorded landslides during the 2001-2018 period. While this, as a matter of fact, is unquestionable, its geomorphic meaning is hard to discuss, in first instance because a single $AR$ value may cover very different 6-week long time series of daily rainfall, from more or less continuous moderate- to high-intensity rainfall over weeks causing deep rotational landslides to very high-intensity rainfall of short duration just before the occurrence of extended shallow landsliding and debris flow. We also observed that a significant percentage (~40%) of the landslide events did not occur on the day when highest rainfall was recorded but one or two days later. As it seems unlikely that all of these landslides would have been wrongly dated, this fact might betray a particular hydrological behaviour of slopes in this tropical environment. Meaningful hypotheses about the interplay between slope physics and rainfall characteristics in this setting will however require in-depth analysis of the 6-week rainfall time series associated with the landslide events. Meanwhile, although this is not straightforward, we can at least attempt a comparison with the results of the many studies based on intensity-duration (ID) or event-duration (ED) analysis of rain gauge data. Extrapolating the ED or ID curves towards a duration of 42 days, many published 5% exceedance probability thresholds fall in the range 75-150 mm over this time length in, e.g., NW Italy (Melillo et al., 2018),



NE Italy (Marra et al., 2017), central Italy (Perucacci et al., 2012; Rossi et al., 2017), Sicily (Gariano et al., 2015), NW USA (Seattle area, Chleborad et al., 2006). Moreover, many landslides that actually occurred after rainfall events of shorter duration were associated with lower cumulated rainfall. The 75-150 mm range is thus an upper bound in these areas and we tentatively suggest an average 50-75 mm cumulated rainfall as representative for antecedent rainfall of landslide events. The

reasons why these figures are still significantly, though not irreducibly, higher than those we obtained in the WEAR are on one hand (1) the fact that most of these studies (except that of Melillo et al., 2018) do not apply a decay function to past rainfall but, on the other hand, might also be related to (2) the very high proneness to sliding on average of weathered material and their interface with the underlying fresh rock on steep slopes of the WEAR, and (3) a poor approximation of SRE underestimation in the WEAR by Monsieurs et al. (2018b) due to the non-linear dependence of underestimation on

rainfall intensity and the weak representativeness of limited gauge data of very local significance with respect to ~28 ×28 lm pixels. Interestingly, in a tropical region similar to the WEAR, namely Papua New Guinea, Robbins (2016) used cumulative rainfall to calculate thresholds based on TMPA data and selected event durations. For an antecedent time length of 40 days, she derived thresholds amounting to ~25 and ~175 mm for short- and long-duration landslide events, respectively. Taking into account that no decay function was involved in her antecedent rainfall calculation, these values are fully consistent with

our data, where short-duration events of shallow landsliding probably determine the 5% threshold of ~9-22 mm (uncorrected for SRE underestimation) in the WEAR whereas long-duration events triggering larger and deeper landslides would make the bulk of noisy high-*AR* (~40-120 mm) data points. Likewise, 5% thresholds estimated in central Italy by Rossi et al. (2017) based on SRE data are in the order of 30 mm cumulative rainfall over an extrapolated duration of 42 days, again fairly similar to our uncorrected ~9-22 mm 5 % thresholds if we take account of the absence of decay function in their

calculations. We also note that our *AR* values are in the range of observed values compiled by Bogaard and Greco (2018), while Guzzetti et al. (2007) even reported extreme values as low as <10 mm. However, Bogaard and Greco (2018) point to the difficulty of interpreting long-duration rainfall measures in terms of average rainfall intensity and their trigger role for shallow landslides and debris flows.

Improvements of our results may be expected in the near future from more regionally-focused susceptibility maps and higher resolution SRE coming soon with the IMERG product, which shows better performance for rainfall detection (Gebregiorgis et al., 2018; Xu et al., 2017). A larger database of dated landslide events would also allow threshold validation, the distinction of landslide types and, thus, the calculation of adapted thresholds.

**6 Conclusion**

In this study, we propose a new rainfall threshold approach fundamentally different from previous research and based on the relation between antecedent rainfall and landslide susceptibility through a modified frequentist approach with bootstrapping. This method has the main advantage of directly mappable susceptibility-dependent rainfall thresholds. Six-week long



antecedent rainfall is calculated based on satellite rainfall estimates from TMPA 3B42 RT. It uses an exponential filter function with a time constant scaled by a power of daily rainfall accounting for the dependence on rainfall intensity of the decaying effect of rain water in the soil. Susceptibility data comes from a study by Broeckx et al. (2018) based on logistic regression and a continental-scale data set of landslides in Africa. Using this method, we identify the first rainfall thresholds

for landsliding in the western branch of the East African Rift, based on a landslide inventory of 143 landslide events over the 2001-2018 period. The obtained *AR* thresholds are physically meaningful and range, without correction for SRE underestimation, from 9.5 mm for the most susceptible areas of the WEAR ($S = 0.97$) to 23.1 mm in the least susceptible areas ($S = 0.38$) where landslides have been reported, for an exceedance probability of 5%. We conclude that the proposed new threshold approach forms an added value to the landslide scientific community, while future improvements are expected

from applying the method to larger data sets and using satellite rainfall estimates with higher spatial (and temporal) resolution and increased rain detection efficiency.

*Author contributions*

AD conceived the new aspects of the method, with input from EM and OD to its development. EM collected the data,

implemented the source code and made all calculations. AD, EM, and OD contributed to the discussion of the results. EM and AD jointly wrote the manuscript, with contribution from OD. OD coordinated and designed this collaborative study in the frame of the RESIST project.

*Competing interests*

The authors declare that they have no conflict of interest.

*Acknowledgements*

We thank the Investigative Reporting Project Italy research institute and their code developers for publically sharing open-source codes for landslide hazard studies. Special thanks go to our partners at Centre de Recherche en Sciences Naturelles de

Lwiro (DR Congo) and Université Officielle de Bukavu (DR Congo), who facilitated fieldwork in the study area and provided information on landsliding. The authors acknowledge NASA Goddard Earth Sciences Data and Information Services Center for providing full access to the precipitation datasets exploited in this study. Datasets can be accessed at https://disc-beta.gsfc.nasa.gov/. We also thank Jente Broeckx, who provided the landslide susceptibility model for Africa. Financial support came from BELSPO for the RESIST (SR/00/305) and AfReSlide (BR/121/A2/AfReSlide) research

projects (http://resist.africamuseum.be/, http://afreslide.africamuseum.be/), and an F.R.S.-FNRS PhD scholarship for EM.

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





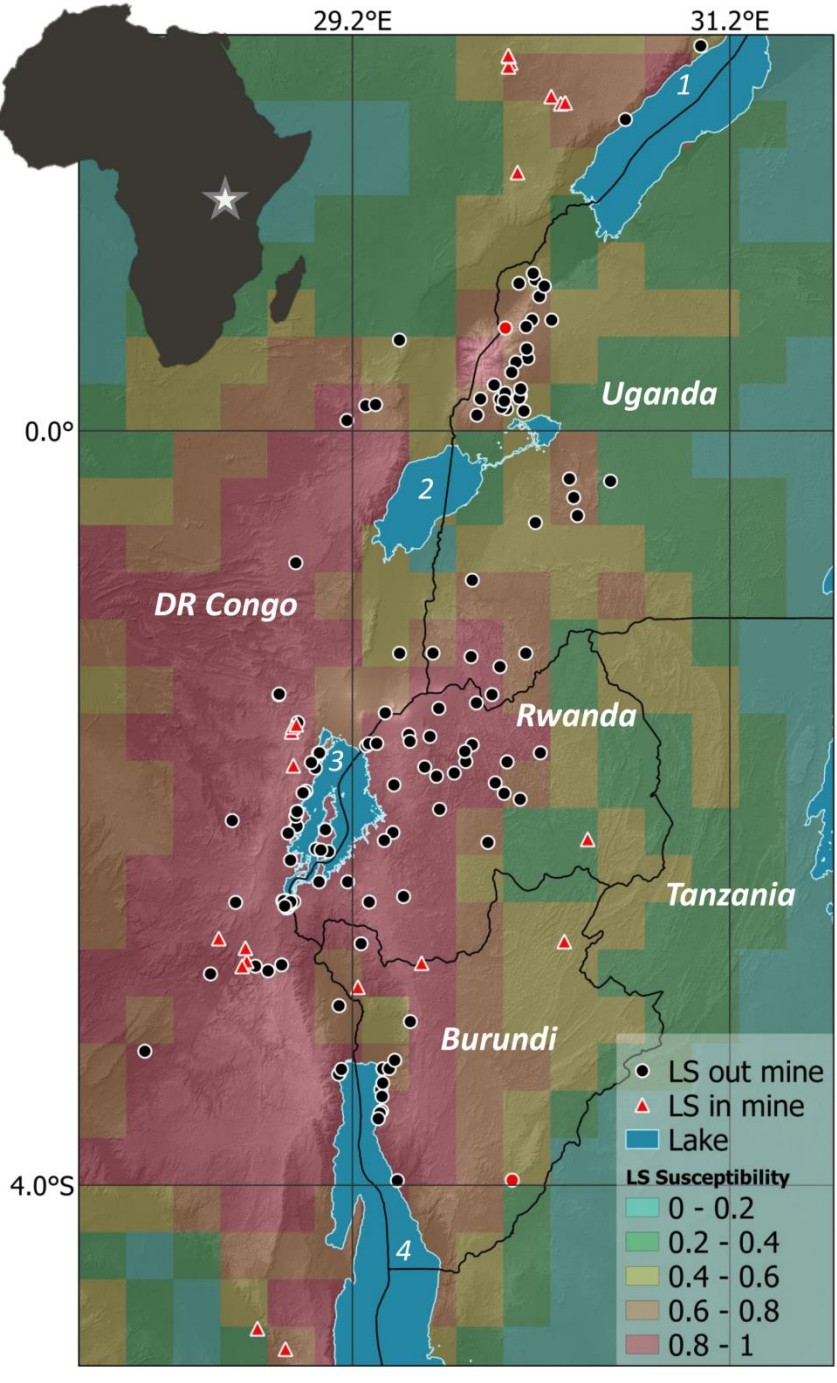

**Figure 1. Landslide susceptibility at 0.25° resolution, derived from the map of Broeckx et al. (2018), and distribution of landslides in the western branch of the East African Rift, comprising 29 landslides in mining areas (triangles), and 145 landslides outside mining areas (dots) of which the red dots are landslides associated with antecedent rainfall less than 5 mm. Only the black dots (143 landslides) are used for calibrating the rainfall thresholds. Numbers in the lakes: 1 = Lake Albert, 2 = Lake Edward, 3 = Lake Kivu, 4 = Lake Tanganyika. Background hillshade SRTM (90m).**




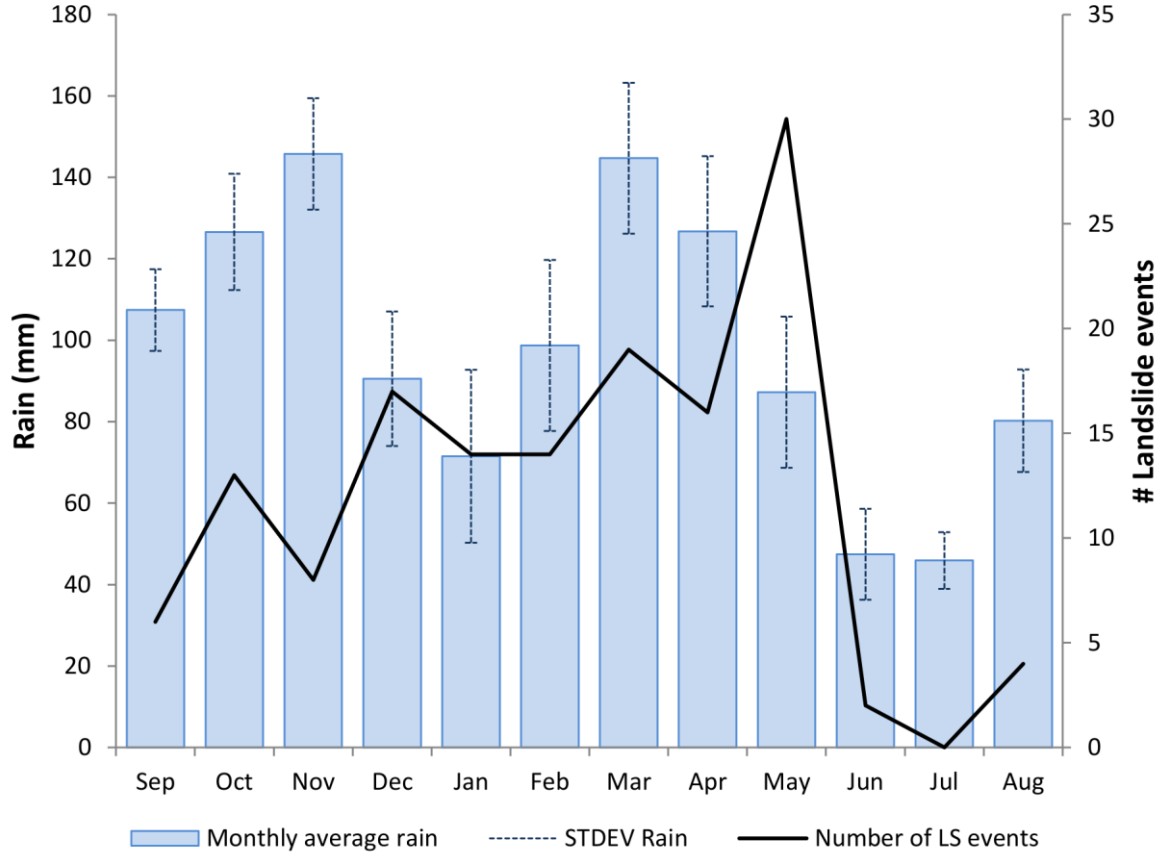

**Figure 2. Monthly distribution of 174 landslide events (LS) in the WEAR and mean monthly rainfall based on 20 years (1998–2018) of TMPA (3B42-RT) daily data, downloaded from http://giovanni.sci.gsfc.nasa.gov.**





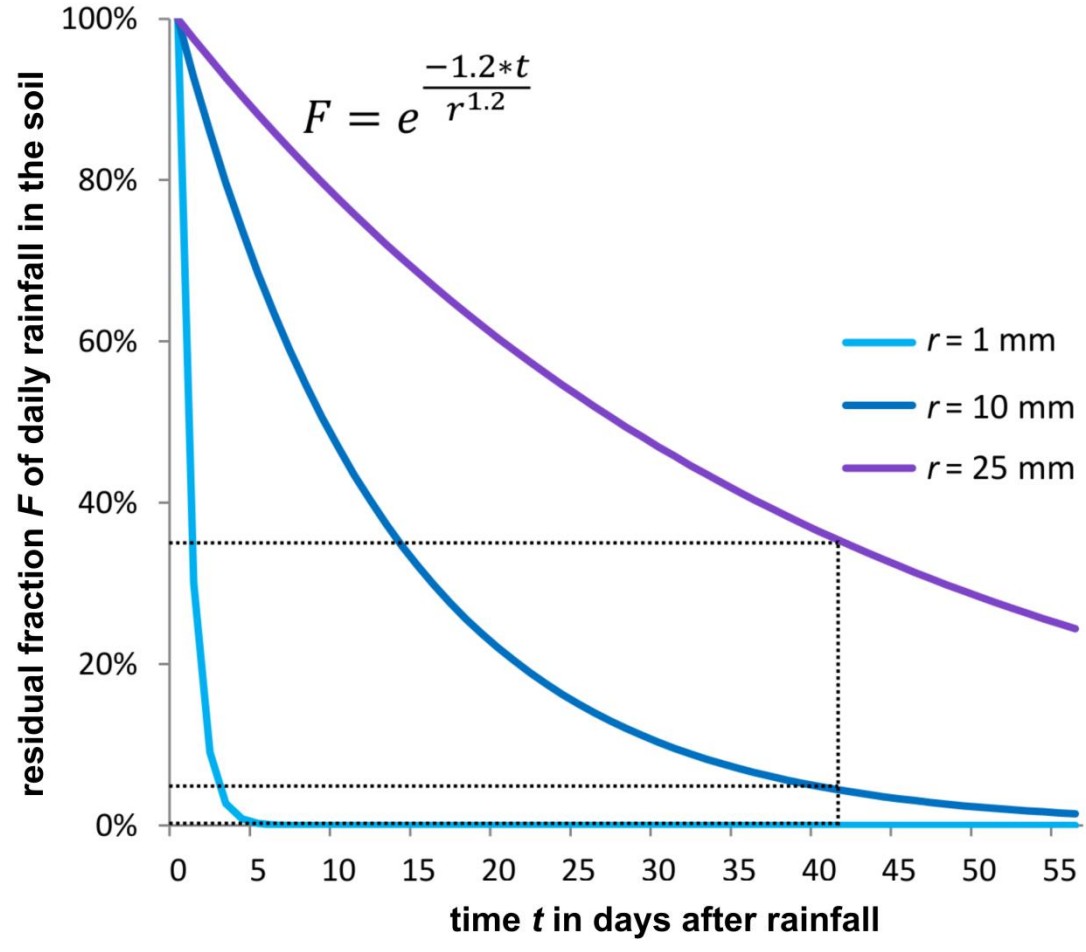

**Figure 3. Decay curves for three daily rainfall of 1 mm, 10 mm, and 25 mm according to the expression of the exponential filter function in equation 2, with *a* = 1.2 and *b* = 1.2. The black dotted lines show that 0%, 4.2%, and 34.7% of the respective original rainfall values are still contributing to the accumulated antecedent rainfall function (Eq. 2) after 42 days (6 weeks).**


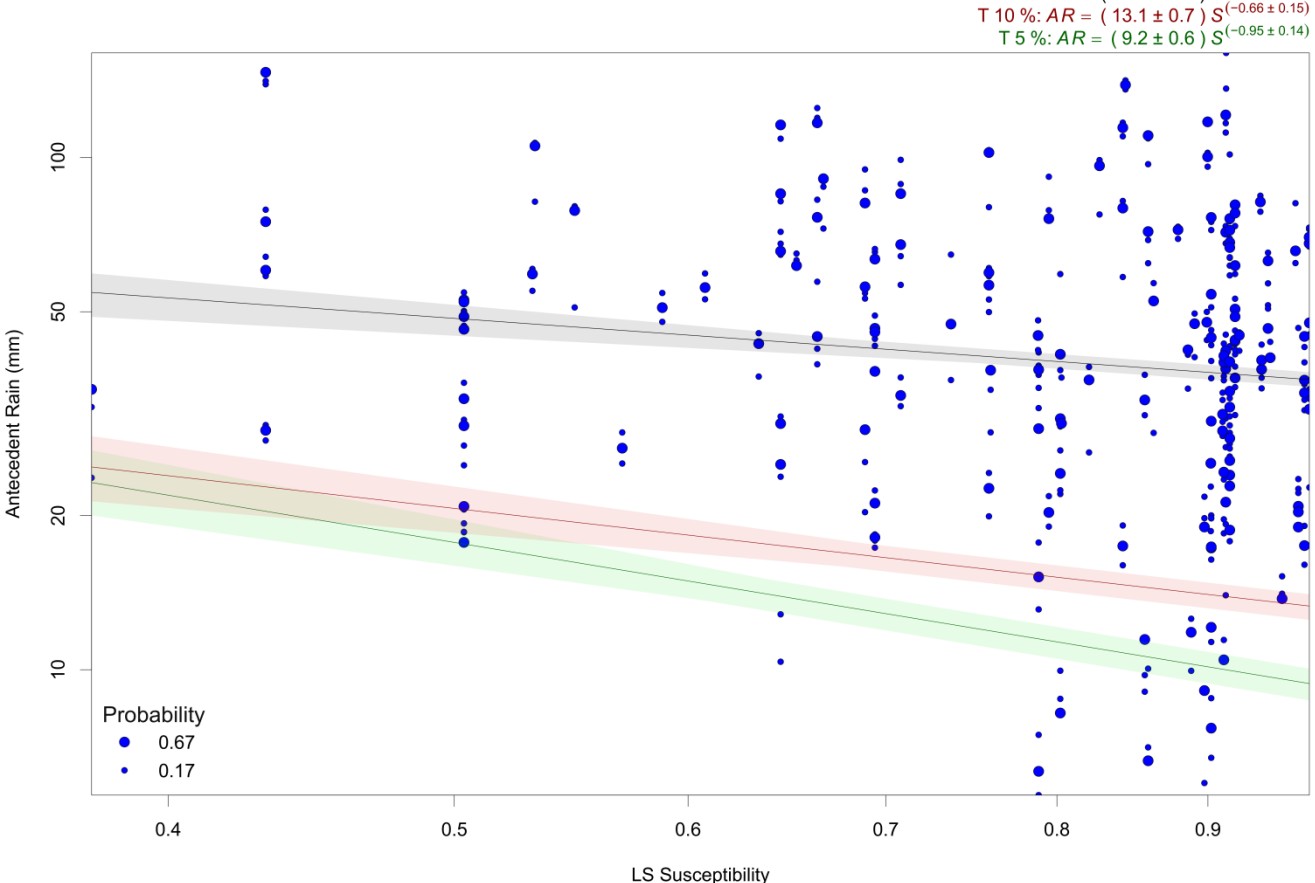

Figure 4. Log-log plot of antecedent rain (mm) *vs* ground susceptibility to landsliding for the recorded landslides, with their associated sampling probability: 0.67 at the reported landslide date; 0.17 at the days prior to and after the reported landslide date. The black curve is the regression curve obtained from the whole dataset; the green and red curves are the *AR* thresholds at 5% and 10% exceedance probability levels respectively, along with their uncertainties shown as shaded areas.





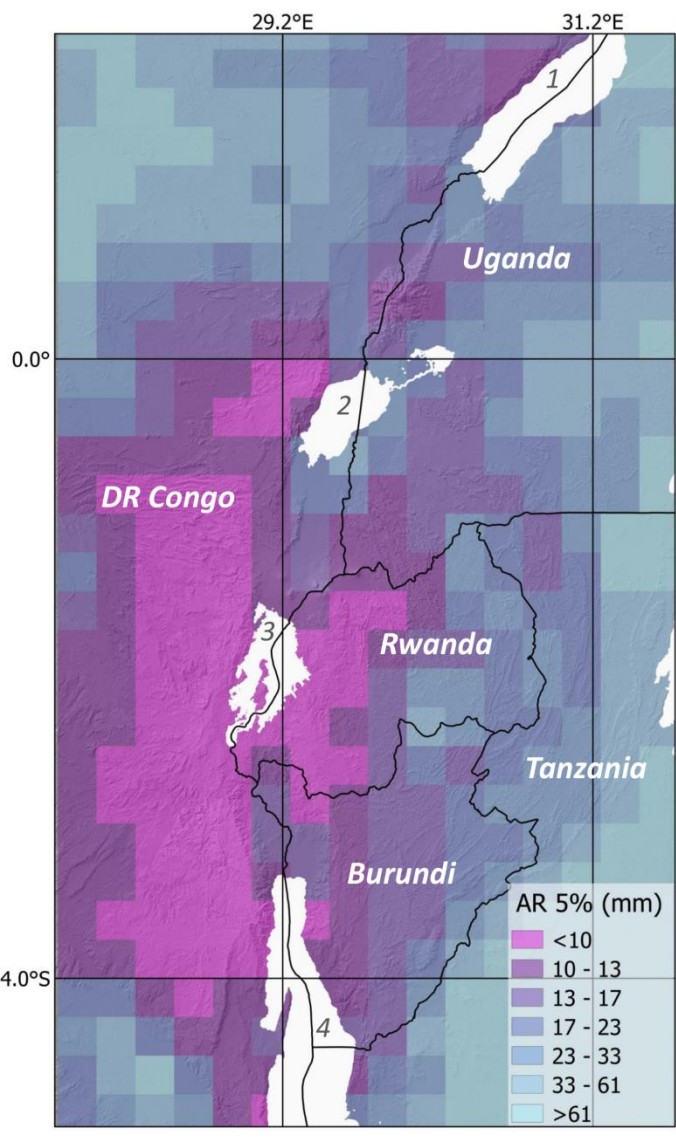

**Figure 5. Antecedent rainfall (*AR*) threshold map (0.25° resolution) at 5% exceedance probability (see Eq. 5). Depending on the local landslide susceptibility (from Broeckx et al. 2018, Fig. 1) threshold values range from *AR* = 9.5 mm in the highest-susceptibility areas (*S* = 0.97) to *AR* = 23.1 mm in the least susceptible pixels (*S* = 0.38) having recorded landslides during the 2001-2018 period. Numbers in the lakes: 1 = Lake Albert, 2 = Lake Edward, 3 = Lake Kivu, 4 = Lake Tanganyika. Background hillshade SRTM (90m).**