# Peer review of "A susceptibility-based rainfall threshold approach for landslide occurrence"

_Natural Hazards and Earth System Sciences, 2018_

## Referee Comment (RC1) · Anonymous Referee #1 · 11 Dec 2018

The main purpose of the paper is to provide a large-scale definition of rainfall threshold for landslide occurrence in the WEAR area using a susceptibility-based approach. The topic is of interest for the scientific community and matches the interest of the NHESS Journal. Significant is also the Authors' attempt to face complex phenomena – like landslides are – trying to use a simple statistical method. Predicting the susceptibility to landslide activity is an important applied problem in natural hazards. The authors rely on previously published models and data. They remind us that the results have to be interpreted carefully given the limitations of the statistical model and data.

On a positive note, the paper is generally well structured and written. Nevertheless, the paper presents some limitations (see comments bellow), concerning both the adopted methodology and some conceptual aspects. For the above reasons, the paper should

be acceptable after minor revisions.

General comments:

A geomorphological map of the study area is missing. It should help readers to better understand the spatial distribution of landslide (Fig.1). The role played by geologic and/or topographic parameters in landslide type and distribution are not clear and must be explained and better addressed by the Authors. A Mean Annual Precipitation map of the study area is although missing.

The methodology developed during this work is clear at all and is written in an intelligible way. Nevertheless, some precision should be given. For example, in Part 2.2: Authors should precise if (1) there are any rain gauges in their study area (maybe a map of the WEAR rain gauges?) and (2) if data of the rain gauges were used in order to perform a calibration and validation process of the TMPA-RT's data.

Specific comments:

Page 2, line 31: Can you please explain and precise what you mean by "the progressive adjustment of landscapes to the governing climatic parameters"?

Page 5, line 12-13: "this model has been produced through logistic regression based on four independent environmental factors, namely topography, lithology, peak ground acceleration and precipitation" . . . I'm not convinced by the fact that these four factors are independent. . . Topography and PGA are strongly related to the lithology, rainfall is strongly related to topography, etc. You should consider to delete "independent".

Page 5, line 16-18: "Interestingly, as their susceptibility map covers the whole Africa, this model characteristic will not contribute to mar potential extrapolations of our calculated thresholds to similar analyses elsewhere in the continent." I do not understand this sentence.

Page 6, line 26: "We empirically determined that a = 1.2 and b = 1.2 provide decay curves that comply with. . ." Please precise how did you "empirically determined" a and

b.

Page 7, line 4-12: Authors should maybe discuss also fast-moving landslides (e.g. debris flows, mudflows...) rainfall thresholds where AR is not a key issue.

Page 10: Authors should give a short title to each subparts (5.1, 5.2, etc.)

Page 10, line 13: The coarse spatial resolution concerns also the susceptibility, some of the controlling factors data cannot be collected in the study area according a thinner resolution. The coarse temporal resolution (lack of hourly rainfall data) can also be a problem, it makes difficult the characterization of the type of landslide.

Page 11, line 9-10: "However, no AR function has so far considered that the decay time constant is likely to increase with rainfall intensity." This is not so clear; it strongly depends on the type of landslide.

Page 12, line 5-7: "(i) probably 5 chiefly, the mixing of all types of landslides in our data set...);" Why authors did not try to make a "raw" classification of the landslide type in their database? Please clarify this point.

---

## Referee Comment (RC2) · Anonymous Referee #2 · 18 Dec 2018

The manuscript proposes a novel threshold for landslide occurrence, based on a non-linear antecedent rainfall index and a landslide susceptibility index (mainly depending on topography and lithology). The porposed threshold is applied to a large area of Eastern Africa, exploting satellite rainfall information, as rain gauge data are lacking in that area.

Such a topic is of interest for the readership of NHESS. The manuscript is well written and organized, and the English language is correct.

However, there are some major issues that should be addressed before the manuscript might be considered for publication.

Specifically, my major concerns are the following:

[Figure]

1. All the elaborations have been carried out without considering AR values which did not correspond to any recorded landslide, although the satellite data used would easily allow it. The authors should explictly mention this choice, explain the reasons for it, and, in the discussion of the results, try to figure out what would be the effects of the inclusion of non-lndslide AR in their calculations.

2. I don't agree with the intepretation of the roles of the variables S and AR, used for the definition of the threshold. In fact, S is an index indicating static geomorphological conditions which make a place more prone to landsliding than another (nothing to do with hydrology, at least not directly). On the other hand, AR, extended over a period of 6 weeks, clearly is not related with triggering rainfall, but mostly on the long-term water accumulation in (and drainage from) the system. So, this AR accounts for hydrological processes leading to predisposing conditions, as well as for characteristics of the triggering rainfall event (the last few days in the AR summation).

3. While I fully agree that a limitation of commonly adopted AR indices is their linearity (i.e., water accumulates always in the same way, regardless of the wetness state of the system), and that the proposed non-linear exponent is a smart way to introduce non-linearity, I disagree with the simplistic intepretation (more rain, longer residence time), which is contradicted by many well-established reuslts of hillslope hydrology, indicating that the wetter a slope is, the faster is the (subsurface) drainage out of it. Hence, i would be more cautious in the discussion of the meaning of the obtained parameter accounting for the non-linearity.

In the attached annotated pdf file, you can find several more detailed comments, which I hope con be of help for the authors to understand my comments, and maybe improve the manuscript.

Please also note the supplement to this comment:
https://www.nat-hazards-earth-syst-sci-discuss.net/nhess-2018-316/nhess-2018-316-RC2-supplement.pdf

**Supplement:**

[revised manuscript text omitted]

---

## Author Comment (AC1) · 21 Dec 2018

This reply is aimed at thanking the reviewer for their work and acknowledging the corresponding need for revision. Specific responses to individual comments will be uploaded with the revised version of the manuscript in the first half of January 2019.

We thank reviewer 1 for his/her time and constructive suggestions. We welcome the comments and will modify/expand the text relative to the reviewer's suggestions where more details are required in order to make our message more clear and comprehensible. We note that the key goal of this paper is the development of the newly proposed threshold methodology, whereas the landslide and rain gauge spatial distributions have mainly been described in the cited previous papers (Monsieurs et al. 2018a,b). We

consider however to add a mean annual precipitation and topographic map as figure or supplementary material.

---

## Author Comment (AC2) · 21 Dec 2018

This reply is aimed at thanking the reviewer for his/her work and acknowledging the corresponding need for revision. Specific responses to individual comments will be uploaded with the revised version of the manuscript in the first half of January 2019.

We thank reviewer 2 for his/her rigorous work and feedback, they make a meaningful contribution to the improvement of our manuscript.

We acknowledge the importance of information on non-landsliding conditions for a comprehensive understanding of the meaning of thresholds for landslide occurrence. As proposed by the reviewer we will explain our choice of the applied approach more explicitly (compared to what is currently described on page 4 L25-28) and discuss the

implications in the discussion.

We appreciate the critical note on the core concept of the paper related to the "trigger-cause framework" adopted for the threshold definition using an antecedent rainfall (AR) – landslide susceptibility (S) relation. The reviewer's insights on the complex relationship between landslide predisposing factors, comprised in the susceptibility index, and soil hydrology status, to be distinguished from the dynamics of soil hydrology, are helpful and will be used to better frame our threshold approach in the revised paper.

The insufficient emphasis put on the non-linear character of the proposed AR function and the latter's link with that part of rainfall that may be identified as the true trigger of landsliding are further valid points and we will make sure to address them in the revised manuscript.

The specific suggestions are helpful in pointing out which concepts should be elaborated in order to clarify or support our message.

———————————————————

---

## Author Response (AR1)

**Nat. Hazards Earth Syst. Sci. Discuss., https://doi.org/10.5194/nhess-2018-316**
A susceptibility-based rainfall threshold approach for landslide occurrence

The reviewers' comments are given in italic and our replies in roman typesetting. After replying to the reviewers, we present the marked-up version of the manuscript.

**Reply to Referee #1**

*The main purpose of the paper is to provide a large-scale definition of rainfall threshold for landslide occurrence in the WEAR area using a susceptibility-based approach. The topic is of interest for the scientific community and matches the interest of the NHESS Journal. Significant is also the Authors' attempt to face complex phenomena – like landslides are – trying to use a simple statistical method. Predicting the susceptibility to landslide activity is an important applied problem in natural hazards. The authors rely on previously published models and data. They remind us that the results have to be interpreted carefully given the limitations of the statistical model and data.*

*On a positive note, the paper is generally well structured and written. Nevertheless, the paper presents some limitations (see comments bellow), concerning both the adopted methodology and some conceptual aspects. For the above reasons, the paper should be acceptable after minor revisions.*

We thank reviewer 1 for his/her time in reviewing the paper.

**General comments**

*A geomorphological map of the study area is missing. It should help readers to better understand the spatial distribution of landslide (Fig.1). The role played by geologic and/or topographic parameters in landslide type and distribution are not clear and must be explained and better addressed by the Authors. A Mean Annual Precipitation map of the study area is although missing.*

Here, the reviewer is looking for understanding the spatial variation in landslide occurrence through predisposing factors such as geology and topography. Beyond limited field observation, we have so far no regional study of the link between predisposing factors and landslide occurrence (this is in progress in parallel by somebody else) and we thus use a continental susceptibility map available for the area. We know it is of limited relevance for a regional threshold approach but it suffices for demonstrating the principle of the method.

We agree that a mean annual precipitation map of the study area in the context of rainfall thresholds brings an added value to the paper, thank you for the suggestion. We add the following text on page 5: "Based on 18 years (2000–2018) of TMPA-RT data, Fig. 1 shows the spatial pattern of mean annual precipitation in the study area, which results from a complex system of climate drivers in equatorial Africa (Dezfuli, 2017).", and the mean annual precipitation map is merged with Fig. 1.

*The methodology developed during this work is clear at all and is written in an intelligible way. Nevertheless, some precision should be given. For example, in Part 2.2: Authors should precise if (1) there are any rain gauges in their study area (maybe a map of the WEAR rain gauges?) and (2) if data of the rain gauges were used in order to perform a calibration and validation process of the TMPA-RT's data.*

Rain gauges in the study area have been used to validate satellite rainfall estimates (TMPA), which has been extensively described in Monsieurs et al. (2018b), cited in the text. The validation results allow us to calculate thresholds from satellite data, however with caution because the limited (in time and space) gauge data did not allow for their robust calibration.

**Specific comments**

*Page 2, line 31: Can you please explain and precise what you mean by "the progressive adjustment of landscapes to the governing climatic parameters"?*

We hereby mean that landscapes tend to a dynamic equilibrium (erosion/deposition) with the governing climate. We believe this concept is known to readers from NHESS and the cited papers of Ritter (1988) and Parker et al. (2016) might bring additional insights if necessary.

*Page 5, line 12-13: "this model has been produced through logistic regression based on four independent environmental factors, namely topography, lithology, peak ground acceleration and precipitation" …I'm not convinced by the fact that these four factors are independent… Topography and PGA are strongly related to the lithology, rainfall is strongly related to topography, etc. You should consider to delete "independent".*

We agree and deleted 'independent'. This revision allowed us to correct the erroneous inclusion of precipitation in the set of environmental factors; precipitation was only included in the logistic regression model calibrated for landslides excluding rockfalls whereas we used the model calibrated for all landslide types (Eq. 3 and 2, respectively, in Broeckx et al., 2018).

*Page 5, line 16-18: "Interestingly, as their susceptibility map covers the whole Africa, this model characteristic will not contribute to mar potential extrapolations of our calculated thresholds to similar analyses elsewhere in the continent." I do not understand this sentence.*

Landslide susceptibility values present probability values with no quantitative meaning. The scale of these probabilities depends strongly on the landslide/no landslide ratio applied in the logistic model calibration. Since the model output covers the entire African continent, we can adopt and compare over the whole Africa our thresholds that have been calibrated with this susceptibility model without problems of differences in the probability scaling of the susceptibility values. We added an additional explanation to the text along with a reference (King and Zeng, 2001) to guide readers who want to gain more insights in this matter.

*Page 6, line 26: "We empirically determined that a = 1.2 and b = 1.2 provide decay curves that comply with…" Please precise how did you "empirically determined" a and b. Page 7, line 4-12: Authors should maybe discuss also fast-moving landslides (e.g. debris flows, mudflows…) rainfall thresholds where AR is not a key issue.*

Values for a and b have been empirically tuned for decay curves to contrast small vs. intense rainfall events reflecting a simplified hypothesis that large rainfall amounts are retained for a longer time in the soil compared to small rainfalls. We acknowledge on page 13 that "A dedicated statistical study of their best values (e.g., Stewart and McDonnell, 1991; McGuire et al., 2002) might perhaps improve somewhat those we empirically defined but, in any case, tests have shown that our formulation of AR is not much sensitive to moderate changes in these values". This is something we are currently looking into but is beyond the scope of the present paper.

*Page 10: Authors should give a short title to each subparts (5.1, 5.2, etc.)*

As this is rather a matter of taste, where we prefer to avoid redundancy in titles with regard to the content in the relatively short subparts, we would like to take the liberty to stick to the current way of presenting the Discussion.

*Page 10, line 13: The coarse spatial resolution concerns also the susceptibility, some of the controlling factors data cannot be collected in the study area according a thinner resolution. The coarse temporal resolution (lack of hourly rainfall data) can also be a problem, it makes difficult the characterization of the type of landslide.*

The resolution of Broeckx et al. (2018) landslide susceptibility data is actually very high (0.0033°), but the coarse satellite rainfall estimate data (0.25°) with which the susceptibility data is compared required a resampling of the susceptibility data to this coarser resolution.

We agree that the temporal resolution is indeed an additional constraining factor, although this concerns the landslide inventory rather than the TMPA satellite rainfall estimates, whose native resolution is 3h. With TMPA's successor, i.e., IMERG, the temporal resolution increases to 30' (as well as the spatial resolution). Still, our analysis is constrained to the temporal resolution of the landslide inventory. Reported landslides in the study area rarely contain information on the time of their occurrence.

We add to the text highlighted by the reviewer: ", coarse temporal resolution of the landslide inventory", and under "2.1 landslides in the WEAR" we add: "Information on the timing of the landslide occurrence is rarely reported.".

*Page 11, line 9-10: "However, no AR function has so far considered that the decay time constant is likely to increase with rainfall intensity." This is not so clear; it strongly depends on the type of landslide.*

We repeat the underlying hypothesis of this statement, namely that large rainfall amounts are retained for a longer time in the soil (longer residence time) compared to small rainfalls and we stress that this fact has so far never been implemented in antecedent rainfall indices. Eq. 2 thus presents improvements over previous AR functions in that it takes account of the non-linear dependence of decay rate on rainfall intensity. The balance between rainfall of the day and antecedent rainfall we aimed at in this expression of AR is important precisely because all types of landslides are included in the data set and AR must, and so does, work for each of them.

*Page 12, line 5-7: "(i) probably 5 chiefly, the mixing of all types of landslides in our data set…);" Why authors did not try to make a "raw" classification of the landslide type in their database? Please clarify this point.*

Only in rare cases landslide reports contain information from which the landslide process could be deduced. A raw classification would in this case not bring an added value to the analysis owing to the high level of uncertainty introduced.

**Reply to Referee #2**

*The manuscript proposes a novel threshold for landslide occurrence, based on a nonlinear antecedent rainfall index and a landslide susceptibility index (mainly depending on topography and lithology). The porposed threshold is applied to a large area of Eastern Africa, exploting satellite rainfall information, as rain gauge data are lacking in that area.*

*Such a topic is of interest for the readership of NHESS. The manuscript is well written and organized, and the English language is correct.*

*However, there are some major issues that should be addressed before the manuscript might be considered for publication.*

We thank reviewer 2 for his time in carefully reading the manuscript and his/her constructive comments which were useful in improving the manuscript.

**General comments**
*Specifically, my major concerns are the following:*

*1. All the elaborations have been carried out without considering AR values which did not correspond to any recorded landslide, although the satellite data used would easily allow it. The authors should explictly mention this choice, explain the reasons for it, and, in the discussion of the results, try to figure out what would be the effects of the inclusion of non-lndslide AR in their calculations.*

We thank reviewer 2 for attracting our attention on the necessity of addressing the issue of all no-landslide AR values, and implicitly, that of the meaning of a high number of 'false positives'. It is true that we can easily calculate AR time series (2000-2018, i.e., ~ 6800-days long) for all ~460 pixels of the study area and have a look at the whole AR data set. However, we feel what we could infer from a quantitative analysis of this data set in terms of false positives is not really worth the additional work required and will in any case be less useful, in the frame of this methodological paper, than a qualitative discussion about type I vs type II errors in the context of early warning against landsliding. Therefore, we added a full point on this topic in the discussion (point 5.4 of the revised version).

*2. I don't agree with the intepretation of the roles of the variables S and AR, used for the definition of the threshold. In fact, S is an index indicating static geomorphological conditions which make a place more prone to landsliding than another (nothing to do with hydrology, at least not directly). On the other hand, AR, extended over a period of 6 weeks, clearly is not related with triggering rainfall, but mostly on the long-term water accumulation in (and drainage from) the system. So, this AR accounts for hydrological processes leading to predisposing conditions, as well as for characteristics of the triggering rainfall event (the last few days in the AR summation).*

We appreciate reviewer 2 for this critical note. The distinction between "trigger" and "cause" remains to date a subject of discussion, e.g.,
- Bogaard & Greco (2015): "A trigger is the last push for a slope to become unstable, whereas the cause is the underlying, often long term, change that occurred preparing the slope for failing.";
- Wieczorek (1996) "Landslides can have several causes, including geological, morphological, physical, and human (Alexander 1992; Cruden and Vames, Chap. 3 in this report, p. 70), but only one trigger (Varnes 1978). By definition a trigger is an external stimulus such as intense rainfall, earthquake shaking, volcanic eruption, storm waves, or rapid stream erosion that causes a nearimmediate response in the form of a landslide by rapidly increasing the stresses or by reducing the strength of slope materials.".

We agree that the interpretation of Bogaard & Greco (2015, 2018) of the landslide "cause" is a *dynamic* hydrological condition representing the soil wetness. Instead, our interpretation lies closer to that of Wieczorek (1996) and we acknowledge that, rather than opposing cause and trigger, the *S − AR* couple refers to static determining ground conditions *versus* dynamic (temporally changing) rainfall conditions that lump trigger and cause *sensu* Bogaard and Greco. The debate rests in fact on the timescales one assigns to trigger and cause. We have clarified in the text that our interpretation slightly differs from the trigger-cause framework of Bogaard and Greco (2018): "Said otherwise, we substitute for the 'trigger-cause' framework proposed by Bogaard and Greco (2018) the coupling of a dynamic meteorologically-based variable ("trigger") and a static indicator of the spatially-varying predisposing ground conditions ("cause")."

*3. While I fully agree that a limitation of commonly adopted AR indices is their linearity (i.e., water accumulates always in the same way, regardless of the wetness state of the system), and that the proposed non-linear exponent is a smart way to introduce nonlinearity, I disagree with the simplistic intepretation (more rain, longer residence time), which is contradicted by many well-established reuslts of hillslope hydrology, indicating that the wetter a slope is, the faster is the (subsurface) drainage out of it. Hence, I would be more cautious in the discussion of the meaning of the obtained parameter accounting for the non-linearity.*

We thank the reviewer for his insights in the matter. We fully acknowledge that our justification of the introduced non-linear exponent was simplistic and thus follow the reviewer's suggestion of only stressing the nonlinearity introduced with $r_k^b$ in the weight function.

**Specific comments**
*In the attached annotated pdf file, you can find several more detailed comments, which I hope con be of help for the authors to understand my comments, and maybe improve the manuscript.*

We thank reviewer 2 for these detailed comments which have been used to improve the paper.

*P2L9-15: The effects of AR on slope stability, as well as the time interval to be considered as representative of antecedent rainfall,is quite different if deep-seated or shallow landslides are considered. When the authors write "AR physical relation with soil shear strength" and "decaying effect of rainfall on soil moisture status", it might seem that the ofcus is on shallow landslides.In any case, it should be stated more clearly if they are referring to any kind of landslide, and, if so, it would be worth marking a distintion between shallow and deep-seated landslides.*

We agree with the reviewer that the effect of AR on slope stability depends on the type of landslide process. The statements *"AR physical relation with soil shear strength"* and *"decaying effect of rainfall on soil moisture status"* are however general and applicable for all landslide types but at varying degrees depending on process (e.g., deep-seated vs shallow). We added this nuance on landslide types into the revised paper.

*P5L16: The meaning of the L/NL ratio, and the way it is used in the susceptibility mapping is unclear and should be better described*

Landslide susceptibility values are probability values with no absolute meaning. These probabilities actually scale with the relative frequency of events (landslide/no landslide ratio) used in the logistic

model calibration. We added a few words to explain this, along with a reference (King and Zeng, 2001) to guide readers who want to gain more insight in this matter.

*P5L29: According to the cited reference, bogaard and Greco (2018), The hydrological cause is a dynamic process, not a "status".*

A particular value of the "hydrological status" refers to a particular time (day) and this status continuously changes, being thus clearly dynamic. This is just a matter of words and we therefore do not agree with the reviewer's comment.

*P5L30-31: The proposed AR extending over six weeks does not refer only to the triggering event. It also accounts for past precipitation, and, given the very long window and the introduced decay, for drainage processes. The suceptibility index is a static factor, which does not take into account of the dynamic predisposing causes.Indeed, I would describe the chosen threshold as a metweorological-geonorphoclimatic threshold.*

We thank the reviewer for his constructive comment. As described under point 2 of the general comments, we mention now clearly our way of interpreting the distinction between trigger and cause; "trigger" presenting a dynamic meteorologically-based variable and "cause" the static predisposing conditions that vary in space.

*P6L1: Again, the concept of triggering does not seem appropriate for the AR variable used here.*

We argue and maintain that the entire six-week period involved in the proposed AR function determines the progressive building up of the landslide "trigger" (see previous response).

*P6L3-4: the adopted susceptibility index has nothing to do with hydrology*

Landslide susceptibility indeed does not reflect the slope hydrology, which we don't claim in our paper. We nevertheless deleted the contentious sentence.

*P6L18-19: Here, rather than trying to interpret the effect of the exponent in terms of residence time, I would stress that a major limitation of the traditional AR definition is that it is linear, while the repsonse of slopes is higly non linear (i.e. the same rainfall record produces different effects in a slope in different initial conditions). So, what the authors are proposing is a non-linear definition of AR*

We thank the reviewer for highlighting the importance of the nonlinear response of slopes to rainfall. There is however a difference between what he seems to mean (the effect of prior soil moisture on how a given rainfall will affect the soil, if we understand him correctly) and what we have in mind (for a given soil status, the fact that a larger rainfall infiltrates deeper and has chances to remain a longer time in the soil than a small rainfall that evaporates (quasi) as fast as it percolates) and we acknowledge again that our view was simplistic because both views should actually be held together. By contrast, if this statement of the reviewer that "*the same rainfall record produces different effects in a slope in different initial conditions*" rather refers to the boundary conditions, then it corresponds to the very essence of our approach combining susceptibility and rainfall characteristics to define susceptibility-, and thus slope-, dependent thresholds but this has nothing to do with any parameter or variable included in the *AR* expression. In any case, as already stated, rephrasing has now switched the general focus from residence time to the (non)linear character of *AR*.

*P6L20-21: This idea should be better discussed and clarified. Indeed, from hillslope hydrology, it is well-known that soil moisture increases the effectiveness of hillslope drainage processes. Hence, it is not as obvious as the authors seem to argue, that "residence time" should increase in rainy days.*

We agree as far as one refers to percentages of total rainfall being rapidly drained but, in absolute terms, the quantity of water remaining in the soil from a large rainfall will most frequently be higher than that from a small rainfall, even though a higher percentage has been drained. We therefore modified the text as follow: "Yet, one may expect that, even though higher percentages are drained for larger rainfall (Dunne and Dietrich, 1980), the quantity of water infiltrating deeper and remaining in the soil from a large rainfall will most frequently be higher than that from a small rainfall. Observation that interception by the canopy, transpiration and evaporation rapidly increase with diminishing rainfall intensity, especially in equatorial areas, also supports this assumption (Schellekens et al., 2000)."

*P6L26-29: A description of how the values of a and b have been chosen should be given. By the way,the cited reference refers to Pennsylvania and not to Africa, and it is not clear what the author mean with "duration of their effect on soil moisture expected in the WEAR".*

Values for a and b have been empirically tuned for decay curves to contrast small vs. intense rainfall events reflecting the hypothesis that decay time is to some extent proportionate to rainfall amount (see responses above). We acknowledge on page 13 that "A dedicated statistical study of their best values (e.g., Stewart and McDonnell, 1991; McGuire et al., 2002) might perhaps improve somewhat those we empirically defined but, in any case, tests have shown that our formulation of AR is not much sensitive to moderate changes in these values". This is something we are currently looking into but developments about how to get best fit parameters are beyond the scope of the present paper. The lack of related scientific research in Central Africa forces us to look at studies done elsewhere, in this case on a ridge system in Pennsylvania, where they found mean residence times of about two months depending on the model used. We therefore reason that the presented fractions of daily rainfall retained in AR after 1.5 months for varying rainfall intensities (Fig. 3) agree with the fairly long mean residence times found in the study of McGuire et al. (2002).

*P7L4-10: This is really a mix-up of very different things. The response time of deep large-scale, slides usually is related to groundwater response to precipitations (in some cases several months), while the cited residence time of two months refers to shallow soil moisture. Finally,the effects of rainfall on creep rate along the San Andreas fault, which is probably affected by preferential infiltration through the fault.*

We are aware of the limitations of these comparisons, called upon for lack of more relevant information. We nevertheless believe that, despite hugely different settings, they provide hints in support of our choice of residence time in shallow and deeper soils. In fact, we just try here to check whether this time length is generally reasonable, without consideration of any particular landslide process. We thank the reviewer for his critical note on the cited study of Roeloffs (2001), which we inserted in the revised manuscript.

*P7L4: This time window definitely indicates that the focus of the study is not in the triggering rain events (or, at least, not only).*

Consistent with our previous responses, we agree with this statement as far as the triggering rain is considered to be the intense daily rainfall just preceding the landslide.

*P7L32: In view of the obtained results (see comment below), it would probably worth some more information about the other tested relationships, and how they performed compared to the power-law*

We agree. We developed this further under '4. AR threshold estimates' (see reply to comment P9L32).

*P8L4-5: This procedure should be described more clearly. Also the cited reference Peruccacci et al. (2012) does not give more information in this respect.*

The bootstrap statistical technique is nothing more than described in the text, i.e., running a model X times using random sampling (we added: "with replacement") for the selection of events to allow the determination of the model uncertainties. We added a reference to the work of Efron (1979) who was the first to introduce this statistical method.

*P8L34: It is most meanningful as long as no evaluation about the false alarms is made*

We added a discussion on the type I and type II errors to this regard (point 5.4 of the Discussion).

*P9L6-8: The non parametric approach followed here requires a rich dataset, so to have a subset large enough to estimate the mean and the standard deviation for a chosen probability.*

We agree. The requirement of a large dataset for this approach has been acknowledged already at different places throughout the text (P2L27; P8L24) and therefore we don't find it necessary to repeat this here.

*P9L15: If a statistical test about the significance of the trend has been carried out, it should be described somewhere*

We added the requested information.

*P9L22: This result would be probably very different (much smaller, I expect) if the missing alarms had been taken into account. I would not draw conclusions about the effects of triggering rainfall by looking only at the rainfall events which actually triggered landslides.*
*Do the authors think that there is 95% probability of triggering if a dot is above the threshold line? So, no false alarms are expected?*

No. This sentence refers to the missed alarms (i.e., false negatives or landslides having occurred for AR below the calculated threshold), thus meaning there is 95% probability for any real event to have occurred for *AR* > threshold. Instead, the problem the reviewer mentions here is that of false alarms, which indeed are not taken into account. We added a discussion on this matter under point 5.4 of the Discussion (see also reply to point 1 of the general comments).

*P9L24-25: Are these trends statistically significant?*

Yes, for almost every bootstrap iteration the parameters of the regression α and β were significant, this is mentioned in the text.

*P9L24-25: I saw in the supplement material that, in the R code used for the trend detemrination, the "general trend" ahs been identified with a probability of 50%. Why not declaring this level of probability?*

Indeed, in the R code we identify the general trend with '50' for an easy distinction of the different variables used throughout the code. The general trend is a least square fit that runs through the middle of the point cloud and hence represents the 50% probability of exceedance threshold. In the text we prefer to refer to this fit as the 'general trend' rather than assigning the level of exceedance probability to it as it is not our purpose to use it as a threshold; it is only used for the calculation of the residuals from this trend in order to define the subsets on which the 5% and 10% probability of exceedance thresholds are calculated.

*P9L28-29: Of course the higher standard deviations are due to the smaller sample size, as the presence of possible replicate data in the randomly extracted sample sizes has more impact if the sample has less data.*

We agree. We deleted 'probably' from this sentence.

*P9L32: This seems just a sophisticated way to say that the lower envelope of the data is more inclined than the median trend..*

Not only is it more inclined (higher β value), but the regression also has an increased fit despite the reduced subset of data points compared to the general trend, which is reflected in the higher values of the determination coefficients $R^2$. We therefore prefer the current phrasing in order to present the complete message we want to bring.

*P9L32: However, the R2 values result quite small, possibly oindicating that the trend is not well described by the chosen power-law equation. Indeed, for both the adopted variables the range of variation is quite limited, if compared with those of the variables usually adopted for the identification of landslide thresholds(especially for the susceptibility index). Are the authors sure that the power law is the best choice for the functional form of the threshold?*

This is a good point. We tested linear, exponential and power fits to the 10% and 20% lowest AR data points and found no significant difference between the determination coefficients of the respective fits, yet the power-law having overall a higher value. With the limited amount of data available (20% of the data for the 10% threshold, i.e., 172 data points, and for the 5% threshold 86 data points) and their limited spread, we cannot however make a conclusion about the best form of the equation for the AR-S relation. We added the following in the revised manuscript: "Though fairly small, these $R^2$ values have proved best among not very different linear, exponential, and power law fits. Better coefficients are probably hampered mainly by inhomogeneities in the subset data distribution within the susceptibility range, with very poor information for S < 0.7 (Fig. 4)…"

*P10L11-12: There is no test of the threshold determination. A threshold has been determined, but no validation has been carried out, nor any evaluation of its performance in term of false alarms.*

We smoothed the phrasing. A large database on landslide events is not easily obtained in the context of Central Africa. In order to establish a new threshold approach based on a frequentist method, we were required to use the entire available dataset for the threshold calibration in order to obtain statistically significant results. The proposed method can be validated once new data on landslide events are available. These points have been explained in the manuscript (P8L23-26; P14L27). However, the very small parameter uncertainties issued by the bootstrap procedure are a strong indication that the calibration is reliable. Moreover, the physically meaningful thresholds obtained in the WEAR (elaborated further in the discussion part P13L18-P14L23) are an additional

hint to a succeeded test. With regard to false alarms, we refer to the reply to point 1 of the general comments.

*P10L18: Again, here it should be clearly stated that the focus is on static ground characteristics, and not to hydrological processes dynamically modifying the ground conditions*

We added '(static)' for clarification.

*P10L26-27: I strongly disagree. It seems that the authors argue that the susceptibility index controls in some way the hydrological processes occurring within the slope and eveltually leading to slope failure (infiltration, evapotranspiration and drainage). In reality, the most important factor related to topography is slope inclination (so that less pressure is needed for slope destabilization), while lithology mainly controls soil mechanical properties (friction angle and cohesion), which have nothing to do with hydrology. Of course, also the infiltration process is affecteed by topography and lithology, but with an indirect and non-linear control. For instance, lithology might somehow affect hydraulic conductivity, but what are the effects of hydraulic conductivity on slope equilibrium? does a high conductivity enhance infiltration or drainage? It is very difficult to draw simplistic conclusions about the hydrological meaning of the chosen susceptibility index (if it exists, actually).*
*I would just say that S is a measure of "weaker" slopes.*

We thank the reviewer for elaborating on the complex relations between ground conditions, rainfall infiltration and drainage, and slope failure. We agree that our statement was misleading so we have removed it.

*P10L31-32: I don't think that a different way of evaluating susceptibility would hamper the applicability of the proposed approach. Also the suseptibility index used here is artificially normalized between 0 and 1, and the probabilities estimated from landsliding history indirectly depend on the susceptibility factors that make a place more prone to landsliding than another one.*

Indeed, a different susceptibility model doesn't hamper applying our threshold approach. We just feel it necessary to emphasize that threshold values obtained from different regions where different susceptibility models have been used cannot be directly compared. We reworded the clause in order to be clearer.

*P11L10: As I already commented above, I would prefer to talk about the non-linearit dependence of time decay on moisture state (and so, on past rainfall).*
*AND*
*P11L10: As a daily rainfall record has been used, I would not use the word "intensity", but simply daily rainfall*

We modified this sentence to focus on the nonlinearity highlighted by the reviewer: "However, no *AR* function has so far considered a nonlinear dependence of the decay time constant on daily rainfall and, thus, soil wetting".

*P12L18: Actually, the false positives have been excluded from the elaborations, although, from the TNPA-RT it would be relatively easy to consider also the AR values which did not correspond to any reported landslides*

Indeed. We now expose reasons for this in more detail in a separate section 5.4 (see also reply to point 1 of the general comments).

*P13L5: No mention of such tests has been made in the paper*

As we now state it, these tests correspond to the empirical evaluation of the behaviour of AR when we empirically varied the parameter values. A more detailed evaluation of the sensitivity of AR to its parameters, which is beyond the scope of this paper, is something we are currently looking into.

*P13L29: Maybe also the most frequent landslide type*

We agree. Thank you for this note, we added to this sentence: "…or be related to specific landslide processes (Montgomery et al., 2002; Lollino et al., 2006)". We took the opportunity of this revision to add two references that are relevant in this context.

*P13L33: I think the authors should emphasize, as another added value of their proposed threshold, that ID or ED thresholds extended to events lasting 42 days are in most cases meanoingless from a hydrological point of view (maybe only for some deep-seated landslides such a long time range may be related to the build-up of groundwater pressure rise). In fact, 42 days of rainfall cannot considered a single rainffall event, and, during such a long period, drainage mechanisms from any kind of slope cannot be neglected. So, simply, the extrapolated values of 75-1500mm make no sense, while the antecedent rainfall, with some decay filter function, represent the mix between predisposing cause and triggering rain, which, together with the information about local susceptibility, leads to a more meaningfulr prediction.*

Indeed, we agree on the limited process understanding for some ID thresholds, as highlighted in the study of Bogaard and Greco (2018), which was cited at the end of this paragraph: "However, Bogaard and Greco (2018) point to the difficulty of interpreting long-duration rainfall measures in terms of average rainfall intensity and their trigger role for shallow landslides and debris flows.".

We thank the reviewer for summarizing this added value of our approach (note however that the extrapolated values are 75-150 (not 1500) mm, which is less nonsensical), which we thought to have implicitly presented, but was maybe not sufficiently stressed. We added the following sentence at the end of this paragraph: "To this extent, another added value of our approach lies in the complex decay filter function used in *AR*, which mixes triggering recent rain and predisposing rain of the past weeks in such a way that the index is meaningful for both shallow and deep-seated landsliding."

*P14L25: I would also mention, aa a possible future improvement, the determination of the threshold by accounting for non-landslide rainfall events, too.*

We expanded this paragraph in order to include this idea.

**Technical corrections**

P2L20 "abandoned" has been replaced by "replaced"

**Additional modifications**

We took the opportunity of the revision to make further changes, not requested by the reviewers:

We have improved the mathematical form of Eq. 1 and Eq. 2, to better express that *AR* time series are calculated from daily rainfall time series.

We specify under '3.2 A new antecedent rainfall function' the dimension of, and unit for, parameter *a*.

We modified P8L28 to correctly present the bootstrap procedure.

We added a brief statement on P14L25 about the limitations of the present results in operational terms.

[revised manuscript text omitted]

---

## Author Response (AR3)

**Nat. Hazards Earth Syst. Sci. Discuss., https://doi.org/10.5194/nhess-2018-316**

A susceptibility-based rainfall threshold approach for landslide occurrence

The reviewers' comments are given in italic and our replies in roman typesetting. After replying to the reviewers, we present the marked-up version of the manuscript.

**Reply to Editor**

The manuscript is well written and interesting, and at tis stage needs only some minor revisions, following the minor points raised by the reviewer, and some suggestions provided in the attached file

We thank the editor for his time and his constructive minor comments.

P2L6 The paper of Vessia et al. 2016 on "Mimic expert judgement through automated procedure for selecting rainfall events responsible for shallow landslide: A statistical approach to validation" is indeed a relevant reference to add to this list, thank you for the suggestion.

P2L26 It is not clear which paper is referred to with "Rossi et al. 2012" and therefore not included in this list of references.

P3L1 Sliding implies a specific type of slope movement. I would rather use the general term "failure" here

We agree that the term "sliding" refers to a specific landslide process and modified this sentence accordingly.

P3L6 It is not clear which paper is referred to with "Rossi et al. 2012" and therefore not included in this list of references.

P3L10 The role of weathering processes in the development of slope failures in different settings worldwide was the object of a specific book by the Geological Society of London. You could find there some interesting references for your work.

Complete reference: Calcaterra D. & Parise M. (Eds.), 2010, Weathering as a predisposing factor to slope movements. Geological Society of London, Engineering Geology Special Publication no. 23, 233 pp.

Thank you for referring to this relevant book. We added the reference for Migoń and Alcántara-Ayala, 2008 on P3L11.

P4L27 I would say this is definitely the main bias! However, it is always like this, even in European case studies

We modified the sentence on P4L26-27 to "...and that another important bias in the WEAR dataset highlighted by field observations is the non-recording of many landslide events..."

**P5L3 spatio-temporal**

This has been modified.

P7L28 This is not clear to me: what do you mean by "human alteration"? In the works by Brunetti et al., and Peruccacci et al., all the landslides located near roads or man-made infrastructures were also excluded, because they were considered as influenced by man. Did you follow the same criteria?

Thank you for pointing out the ambiguity in the text. Taking into account the lack of information for most landslide reports in Central Africa and the location inaccuracy, it was not possible to identify and exclude landslides near roads or man-made infrastructures. The text "and human alteration of the ground" is deleted from the text.

**P15L27 These references should be in chronological order**

Throughout the paper we choose to cite references in alphabetical order, following the NHESS guidelines (https://www.natural-hazards-and-earth-system-sciences.net/for\_authors/manuscript\_preparation.html : "In terms of in-text citations, the order can be based on relevance, as well as chronological or alphabetical listing, depending on the author's preference.").

**Reply to Reviewer**

The new version of the manuscript has been substantially improved.

Nearly all the issues I raised have been satsfactorily addressed, so I still have only a minor comment about the newly added Section 5.4, The discussion about the neglected false positives is well argumented. I just comment that, if many FPs occur, although "less than what might seem at first glance" (it would be interesting to know, if you cut the long tail of FPs after a TP, how much the numbers would reduce in your example pixel), it becomes untrue that when the warning is issued there is a high probability of the occurrence of a landslide (this depends on how you estimate your probability, and the many FPs are telling us that the estimate is far from being accurate). So, I would rephrase a little bit this new part. However, this is just a minor remark, and so I leave to the authors the decision to consider some further adjustment or not.

We thank the reviewer for his/her time in carefully reading the paper for a second time.

The reviewer's suggestion to look at the reduced number of FPs when a specified number of days after each TP would be excluded from the AR time series is indeed interesting. A separate analysis should be dedicated to obtain an appropriate period of time that should be excluded from the AR time series after a landslide occurred (P13L5-6 "...the subsequent construction-dependent slow return of the index to below-threshold values frequently last for days or weeks without further landsliding."). This is beyond the scope of the present paper. Moreover, the result might be misleading since other sources of "false false positives" still remain, including the bias of non-reported landslides. We therefore do not develop this further in the current paper, but we will take this into account in our subsequent work.

We agree that the sentence highlighted by the reviewer could be rephrased with more caution. The sentence on P13L10-12 has been modified accordingly.

[revised manuscript text omitted]